# Herbal Combination of *Phyllostachys pubescens* and *Scutellaria baicalensis* Inhibits Adipogenesis and Promotes Browning via AMPK Activation in 3T3-L1 Adipocytes

**DOI:** 10.3390/plants9111422

**Published:** 2020-10-23

**Authors:** Yoon-Young Sung, Eunjung Son, Gayoung Im, Dong-Seon Kim

**Affiliations:** 1Herbal Medicine Research Division, Korea Institute of Oriental Medicine, 1672 Yuseong-daero, Yuseong-gu, Daejeon 34054, Korea; yysung@kiom.re.kr (Y.-Y.S.); ejson@kiom.re.kr (E.S.); 2Nova K Med Co., Ltd., 1646 Yuseong-daero, Yuseong-gu, Daejeon 34054, Korea; imci85@novarex.co.kr

**Keywords:** adipogenesis, AMPK, anti-obesity, triglycerides, UCP1

## Abstract

To investigate the anti-obesity effects and underlying mechanism of BS21, a combination of *Phyllostachys pubescens* leaves and *Scutellaria baicalensis* roots was used to investigate the effects of BS21 on adipogenesis, lipogenesis, and browning in 3T3-L1 adipocytes. The expression of adipocyte-specific genes was observed via Western blot, and the BS21 chemical profile was analyzed using ultra-performance liquid chromatography (UPLC). BS21 treatment inhibited adipocyte differentiation and reduced the expression of the adipogenic proteins peroxisome proliferator-activated receptor γ (PPAR-γ), CCAAT/enhancer-binding protein (C/EBP-α), and adipocyte protein 2 (aP2), as well as the lipogenic proteins sterol regulatory element-binding protein 1c (SREBP-1c) and fatty-acid synthase (FAS). BS21 enhanced protein levels of the beta-oxidation genes carnitine palmitoyltransferase (CPT1) and phospho-acetyl-coA carboxylase (p-ACC). BS21 also induced protein expressions of the browning marker genes PR domain containing 16 (PRDM16), peroxisome proliferator-activated receptor gamma co-activator 1-alpha (PGC1α), and uncoupling protein (UCP) 1, and it induced the expression of the thermogenic gene UCP2. Furthermore, BS21 increased adenosine monophosphate-activated protein kinase (AMPK) activation. UPLC analysis showed that BS21 contains active constituents such as chlorogenic acid, orientin, isoorientin, baicalin, wogonoside, baicalein, tricin, wogonin, and chrysin. Our findings demonstrate that BS21 plays a modulatory role in adipocytes by reducing adipogenesis and lipogenesis, increasing fat oxidation, and inducing browning.

## 1. Introduction

Obesity is the main risk factor for several metabolic abnormalities including hyperlipidemia, atherosclerosis, type 2 diabetes, dyslipidemia, and hypertension [1]. Obesity develops following a prolonged period of positive energy imbalance when food intake exceeds total energy expenditure, which must be reversed for treatment to be successful [2,3]. New approaches for promoting energy consumption are emerging to inhibit energy intake (appetite or absorption) [4]. Brown adipocytes in brown adipose tissue (BAT) expend stored energy and produce heat via the expression of brown fat-specific uncoupling protein (UCP) 1 [5]. Beige adipocytes are inducible brown-like fat cells in white adipose tissue (WAT) that can consume stored energy following exposure to adrenergic signaling or chronic cold through UCP1-mediated thermogenesis, and recent treatments for obesity have increased the therapeutic interest in browning of WAT [6].

Adenosine monophosphate-activated protein kinase (AMPK) is an intracellular energy sensor that regulates energy balance and acts as a metabolic switch [7]. AMPK stimulates adenosine triphosphate (ATP)-producing catabolic pathways (e.g., lipolysis or fatty-acid oxidation) and switches off ATP-consuming anabolic pathways (e.g., lipogenesis) that increase energy production and consumption, respectively [8]. Thus, the development of AMPK activators has become a therapeutic target for obesity [9].

Bamboo leaves, usually as tea, have been used for more than 1000 years in Asia for nutrition and as herbal medicine and have various biological functions, including antibacterial, antiviral, and anti-atherosclerosis activities [10]. The antioxidant and anticoagulant effects of *Phyllostachys pubescens* Mazel (Bamboo, Poaceae family) leaves have been reported [11]. It was recently reported that bamboo leaves and its major flavonoid, isoorientin, inhibited adipogenesis in adipocytes [12,13]. *Scutellaria baicalensis* Georgi (Baikal skullcap, family: Lamiaceae family) roots (S) have been used in traditional medicine. These roots have diuretic, anti-diarrhea, and anti-inflammation effects [14] and were recently reported to reduce body weight and improve serum lipid levels in mice [15,16]. Baicalin in *S. baicalensis* exhibited anti-obesity effects through enhanced fatty-acid oxidation [17]. Our recent study suggested that various mixtures of *P. pubescens* leaves and *S. baicalensis* roots (BS) had anti-obesity effects in obese mice fed a high-fat diet, and a 2:1 (*w*:*w*) mixture of the two plant extracts (BS21) was most effective in decreasing body weight gain and normalizing serum lipid profiles [18]. However, the anti-obesity effects of BS21 and its underlying mechanisms in adipocytes have not been explored. This study examined the effect of BS21—the most effective combination—on adipogenesis and browning in 3T3-L1 adipocytes. Ultra-performance liquid chromatography (UPLC) was performed to determine the constituents of BS21 and assess which constituent inhibited adipogenesis.

## 2. Results

### 2.1. Effect of BS21 on Adipocyte Differentiation and Lipolysis in 3T3-L1 Cells

Preadipocytes were stained with fat-specific Oil Red O after inducing adipocyte differentiation in MDI (3-isobutyl-1-methylxanthine, dexamethasone, and insulin) differentiation media (Figure 1A,B). Oil Red O staining quantification demonstrated that treatment with 60, 120, 240, and 480 µg/mL concentrations of BS21 significantly decreased adipocyte differentiation as well as the accumulation and size of lipid droplets, as shown by a ≈50% decrease in fat accumulation with 240 µg/mL treatment. After adipocyte differentiation, the secreted levels of adipokines (leptin and adiponectin) were measured in the supernatant (Figure 1C). BS21 at concentrations of 240 and 480 µg/mL attenuated an increase in leptin levels following differentiation, and 480 µg/mL also decreased adiponectin levels. The glycerol concentration in culture supernatants was used as a marker for adipocyte lipolysis. As shown as Figure 1D, BS21 did not increase lipolysis. Rather, 400 µg/mL of BS21 decreased glycerol levels. None of these extracts affected cell viability after 24 or 48 h treatment at any concentration tested (Figure 1E).

### 2.2. Effect of BS21 on Adipocyte Marker Protein Expression in 3T3-L1 Cells

We next confirmed the expression of white adipocyte marker proteins in 3T3-L1 cells. BS21 treatment for 7 days significantly reduced the protein expression of white adipocyte markers (peroxisome proliferator-activated receptor γ (PPARγ), CCAAT/enhancer-binding protein (C/EBPα), and adipocyte protein 2 (ap2)) and lipogenic markers (sterol regulatory element-binding protein (SREBP1c) and fatty-acid synthase (FAS)) in a dose-dependent manner (Figure 2A,B). Expression levels of the β-oxidation proteins (CPT1 and p-ACC) were significantly increased by 400 μg/mL BS21. The phosphorylation of the lipolysis gene HSL decreased with BS2. These results suggest that BS21 can regulate lipid metabolism in 3T3-L1 adipocytes.

### 2.3. Effect of BS21 on Browning and AMPK Signaling in 3T3-L1 White Adipocytes

To assess whether BS21 regulates browning, we measured the protein levels of browning marker genes. BS21 at concentrations of 240 and 480 μg/mL induced expressions of specific brown fat proteins UCP1, PRDM16, and PGC1α as well as the thermogenesis-related gene UCP2 (Figure 3A,B).

### 2.4. Effect of BS21 on AMPK Signaling in 3T3-L1 Cells

We next investigated whether BS21 affects lipid metabolism and browning via AMPK. BS21 at concentrations of 240 and 480 μg/mL increased AMPK phosphorylation (Figure 3C). This finding suggested that the increased lipid metabolism and browning effect in BS21-treated 3T3-L1 adipocytes could be mediated by AMPK activation. To further examine the AMPK signaling mechanism of the browning effects of BS21, we treated an AMPK antagonist (dorsomorphin). Elevated UCP1 expression was diminished by treatment with dorsomorphin (Figure 3D). These results indicate that BS21 induced the browning of 3T3-L1 adipocytes by enhancing AMPK phosphorylation.

### 2.5. Effect of Nine Major BS21 Compounds on 3T3-L1 Adipocyte Differentiation

UPLC analysis showed that BS21 contained chlorogenic acid, orientin, baicalin, wogonoside, baicalein, tricin, wogonin, and chrysin (Figure 4 and Table 1). Groups of cells were treated with one of the nine compounds at concentrations of 10 and 50 μg/mL, and all compounds significantly reduced adipocyte differentiation and leptin levels except for wogonoside (Figure 5A–C). In particular, baicalein and baicalin at 10 μg/mL inhibited adipocyte differentiation by 50% compared with the differentiated MDI cells. These compounds did not affect cell viability at all concentrations tested (Figure 5D).

## 3. Discussion

This study offers the first direct evidence on the anti-obesity effects of BS21 achieved through anti-adipogenesis and the promotion of browning. The results also provide insights into the regulatory mechanisms underlying those effects in 3T3-L1 adipocytes.

PPARγ, C/EBPα, and SREBP-1c are major transcription factors involved in adipocyte differentiation, lipid accumulation, and glucose homeostasis [19,20]. These proteins regulate the expression of adipocyte-specific markers such as FAS, adiponectin, leptin, and aP2 [8,21]. In this study, we found that BS21 decreased the protein levels of transcription factors PPARγ (although it increased at 400 μg/mL concentration), C/EBPα, and SREBP-1c as well as their downstream target proteins FAS, aP2, leptin, and adiponectin in 3T3-L1 adipocytes. BS21 did not enhance lipolysis and the protein levels of phospho-hormone-sensitive lipase (p-HSL), which is an important intracellular neutral lipase that degrades triglycerides by hydrolyzing the ester bond [22]. BS21 also increased protein levels of UCP1 and UCP2, which are mainly associated with thermogenesis and energy expenditure in brown and white adipocytes, respectively [23,24]. These results are supported by decreased fat accumulation in BS21-treated 3T3-L1 adipocytes. Collectively, our findings suggest that BS21 inhibited adipocyte differentiation by down-regulating lipogenesis and up-regulating thermogenesis during adipogenesis.

To further elucidate the mechanism underlying the anti-obesity effects of BS21, we measured marker proteins of BAT (PR domain containing 16 (PRDM16), peroxisome proliferator-activated receptor gamma co-activator 1-alpha (PGC1α), and UCP1) involved in adaptive thermogenesis and the transition of WAT to brown-like adipose tissue [25]. In addition, the activated β-adrenergic signaling cascade (protein kinase A, AMPK, p38, and extracellular signal-regulated kinase) induces white fat browning and brown fat development by elevating PGC1α, PRDM16, PPARγ, and UCP1 [26,27,28,29]. In this study, BS21 treatment induced AMPK activation and the expression of brown adipocyte marker proteins in 3T3-L1 adipocytes. In addition, BS21 treatment at 480 ug/mL increased protein levels of PPARγ despite the inhibition of adipogenesis. Previous studies reported that ginsenoside Rb1 and curcumin promote the browning of 3T3-L1 adipocytes through the induction of PPARγ [30,31]. To identify its role in the BS21-induced browning pathway, we treated adipocytes with dorsomorphin, which is a selective inhibitor of AMPK. The arrest of brown fat expression marker protein UCP-1 upon AMPK inhibition reinforced our conclusion that the BS21 mixture induces browning via the AMPK-mediated pathway. Results from previous studies and the present work suggest that BS21 promotes the browning of white adipocytes through the up-regulation of AMPKα-induced white browning proteins PGC1α, PRDM16, and UCP1.

Activated AMPK switches on mitochondrial fatty-acid oxidation and lipolysis [7] and turns off fatty-acid synthesis by inducing the phosphorylation and inactivation of acetyl-CoA carboxylase (ACC), which is an inhibitor of mitochondrial protein CPT1 activity [32]. BS21 treatment led to increased expressions of p-ACC and CPT1. Therefore, the enhanced expression of these proteins is likely to reflect the stimulation of fat oxidation by BS21 treatment. However, even though AMPK activation was increased by BS21, lipolysis and HSL phosphorylation were decreased after BS21 treatment. These data indicate that the anti-obesity effects of BS21 are independent of the HSL enzyme. A recent study showed that AMPK activation inhibited lipolysis [33]. Berberine, a hypoglycemic agent with anti-dyslipidemia and anti-obesity activities, exerts anti-lipolytic effects mainly by reducing the inhibition of phosphodiesterase in 3T3-L1 cells, leading to a decrease in cAMP and HSL phosphorylation independent of the AMPK pathway [34]. These results suggest that the effect of BS21 on AMPK activation may be inhibited or independent of lipolysis.

These results suggest that BS21 may attenuate adipocyte differentiation by suppressing lipid synthesis and promoting fatty-acid oxidation by regulating AMPK activation. UPLC identified nine main compounds (chlorogenic acid, orientin, isoorientin, baicalin, wogonoside, baicalein, tricin, wogonin, and chrysin) in BS21. In our previous study, the two marker compounds isoorientin and baicalin were identified and quantified for the standardization of BS21 [18]. In this study, we identified seven compounds as well as isoorientin and baicalin. Eight constituents (except wogonoside) inhibited adipocyte differentiation and triglyceride accumulation and decreased leptin levels in 3T3-L1 cells. Consistent with our results, previous studies reported that baicalin, baicalein, wogonin, orientin, and isoorientin inhibited adipogenesis in 3T3-L1 adipocytes by decreasing expressions of C/EBPα, SREBP-1c, and PPARγ [13,35,36,37,38]. Chlorogenic acid was previously shown to inhibit adipocyte differentiation via AMPK activation [39,40]. Tricin suppressed the expression of transcription factors associated with adipocyte differentiation (PPARγ and C/EBPα) in obese mice fed a high-fat diet [41]. Collectively, the results suggest that these compounds are responsible for the anti-adipogenic activity of BS21. However, further investigation into the effects of these compounds on obesity is required.

The BS2 extract contains two plant extracts and a variety of compounds. A limitation of this study is that it is unclear if one or multiple components of BS21 are responsible for the observed effects on AMPK phosphorylation and thermogenesis. In addition, whether BS2 and its compounds decrease obesity in vivo using these energy expenditure mechanisms needs further investigation.

## 4. Materials and Methods

### 4.1. Preparation of BS21

*P. pubescens* leaves and *S. baicalensis* roots for the BS21 mixture were purchased from Zhenjiang KOC Biotech Co., Ltd. (Jiangsu, China). The dried *P. pubescens* leaves and *S. baicalensis* roots were extracted with 70% ethanol solution for 3 h; then, the extract was prepared by mixing with a 2:1 ratio of *P. pubescens* (B) and *S. baicalensis* root (S) extracts. The lyophilized powder was dissolved in 10% dimethyl sulfoxide and filtered through a 0.22-μm syringe filter to make the stock solution.

### 4.2. Chemical Profiling of BS21

A Waters ACQUITY UPLC system equipped with a quaternary pump, auto-sampler, and photodiode array detector with an ACQUITY UPLC BEH C18 column (100 × 2.1-mm, 1.7-μm) was used for the chemical profiling of BS21 (Waters Corp., Milford, MA, USA). Elution was performed with 0.1% phosphoric acid solvent A and acetonitrile solvent B. The elution conditions involved holding the starting mobile phase at 94% A and 6% B for 12 min and applying a gradient (89% A: 11% B for 3 min, 80% A: 20% B for 20 min, 40% A: 60% B for 1 min). A wash with 100% B was applied for 2 min, followed by equilibration at 94% A and 6% B for 2 min. The separation temperature was maintained at 40 °C throughout the analysis, with a flow rate of 0.5 mL/min and an injection volume of 3 µL. Each component was analyzed by Quadrupole Dalton (QDa) mass spectrometry, after which the reference standards were purchased from Sigma-Aldrich (St. Louis, MO, USA) and ChemFaces Inc. (Wuhan, Hubei, China), respectively. The retention time and UV spectra were compared and quantified. Components were quantified based on peak areas at 270 nm.

### 4.3. 3T3-L1 Cell Culture and Differentiation

3T3-L1 cells (ATCC, Rockville, MD, USA) were incubated in Dulbecco’s modified Eagle’s medium (DMEM) supplemented with calf serum (10%) and penicillin/streptomycin (100 µg/mL) in a humidified atmosphere of 5% CO_2_ at 37 °C. To induce the differentiation of 3T3-L1 preadipocytes, cells were incubated with DMEM containing 10% fetal bovine serum (FBS), 3-isobutyl-1-methylxanthine (0.5 mM), dexamethasone (1 µM), and insulin (10 µg/mL) (MDI differentiation media) after reaching confluence in 24-well plates. After 2 days of induction, cells were cultured in DMEM containing FBS (10%) and insulin (10 µg/mL) for 2 days and subsequently maintained in DMEM containing FBS (10%). Various concentrations (60, 120, 240, and 480 µg/mL) of BS21 and its constituents (10 and 50 µg/mL) were added along with the differentiation medium. To test the effects of p-AMPK inhibitor (dorsomorphin, Sigma, St. Louis, MO, USA), cells were supplemented with dorsomorphin (5 µM) in differentiation media until harvest. On day 7, differentiation was assessed based on Oil Red O staining and morphological changes. A cytotoxicity test of 3T3-L1 cells was performed to examine the effect of BS21 on 3T3-L1 cell viability using the 3-(4,5-dimethylthiazol-2-yl)-2,5-diphenyltetrazolium bromide (MTT) assay. The experiment was performed in triplicate, and cell viability was calculated relative to that of the BS21 no-treated control cells.

### 4.4. Oil Red O Staining

For Oil Red O staining, the differentiated cells were washed gently with phosphate-buffered saline (PBS) and fixed with 10% formalin for 15 min. The fixed cells were stained with Oil Red O solution (in 60 isopropanol and 40% water) for 30 min. After staining, the lipid droplets were visualized using an Olympus CKX41 microscope (Olympus, Tokyo, Japan) and photographed at 100× magnification using the Motic image Plus 2.0 program (Motic, Hong Kong, China).

### 4.5. Measurement of Lipid Content

To quantify intracellular lipids in Oil Red O-stained cells, the stained lipid droplets were dissolved in isopropanol (500 µL/well) for 30 min, and the extracted dye (100 µL) was transferred to a 96-well plate. Absorbance was measured at 520 nm using a spectrophotometer (BioRad, Hercules, CA, USA). The optical density of the fully differentiated adipocytes was set at 100% relative lipid content.

### 4.6. Measurement of Adipokines

After differentiation, leptin and adiponectin levels in the medium were measured using a leptin mouse enzyme-linked immunosorbent assay (ELISA) kit (Abcam, Cambridge, UK) and adiponectin mouse ELISA kit (R&D systems, Minneapolis, MN, USA), respectively, following the manufacturer’s manuals.

### 4.7. Measurement of Glycerol Content

Glycerol levels in the medium were measured using a glycerol assay kit (Sigma) with glycerol standards for calibration. Briefly, 100 μL of free glycerol reagent reconstituted in distilled water was mixed with 10 μL of the test sample. Thereafter, the mixtures were incubated at room temperature for 20 min, and the solution absorbance was measured at 570 nm using a microplate reader.

### 4.8. Western Blot Analysis

Total protein from cells was extracted with PRO-PREP protein extraction solution (Intron, Seoul, Korea) according to the manufacturer’s instructions. Total protein samples (15 μg) were separated by sodium dodecyl sulfate polyacrylamide gel electrophoresis and transferred onto nitrocellulose membranes. The membranes were blocked with EzBlock Chemi (ATTO, Tokyo, Japan) and then incubated overnight with primary antibodies at 4 °C. Primary antibodies to peroxisome proliferator-activated receptor (PPAR) γ, CCAAT/enhancer-binding protein (C/EBP) α, sterol regulatory element-binding protein (SREBP)-1c, UCP2, and carnitine palmitoyltransferase (CPT) 1 were obtained from Santa Cruz Biotechnology (Dallas, TX, USA). Antibodies to adipocyte protein 2 (aP2), fatty-acid synthase (FAS), UCP1, AMPKα, phospho-AMPKα, and β-actin were purchased from Cell Signaling (Danvers, MA, USA). Antibodies to phospho-hormone-sensitive lipase (HSL), PR domain containing 16 (PRDM16), and PPARγ co-activator 1-alpha (PGC1α) were obtained from Abcam, and antibodies to phospho-Acetyl-CoA carboxylase (p-ACC) were purchased from Millipore (Billerica, MA, USA). After incubation with primary antibodies, the membranes were incubated with the corresponding horseradish peroxidase-conjugated secondary antibodies (Cell Signaling Technology) for 1 h at room temperature. Then, the membranes were treated with ECL chemiluminescent detection reagent (Amersham Bioscience, Little Chalfont, UK), and protein bands were visualized using LAS-1000 (Fujifilm, Tokyo, Japan). The relative band density was determined by densitometry with Image J software (National Institutes of Health, Bethesda, MD, USA).

### 4.9. Statistical Analysis

All experiments were repeated at least three times, and each experiment was performed in triplicate. Differences among experimental groups were evaluated using one-way analysis of variance followed by Dunnett’s multiple-range tests. All data are presented as mean ± standard deviation (SD). For all tests, *p* < 0.05 was considered statistically significant.

## 5. Conclusions

The herbal mixture BS21 exerted anti-obesity effects in 3T3-L1 adipocytes by reducing adipogenesis and activating thermogenesis and browning via AMPK activation. Therefore, the data suggest that BS21 might be a potential therapeutic candidate for the prevention and treatment of obesity.

## Figures and Tables

**Figure 1 plants-09-01422-f001:**
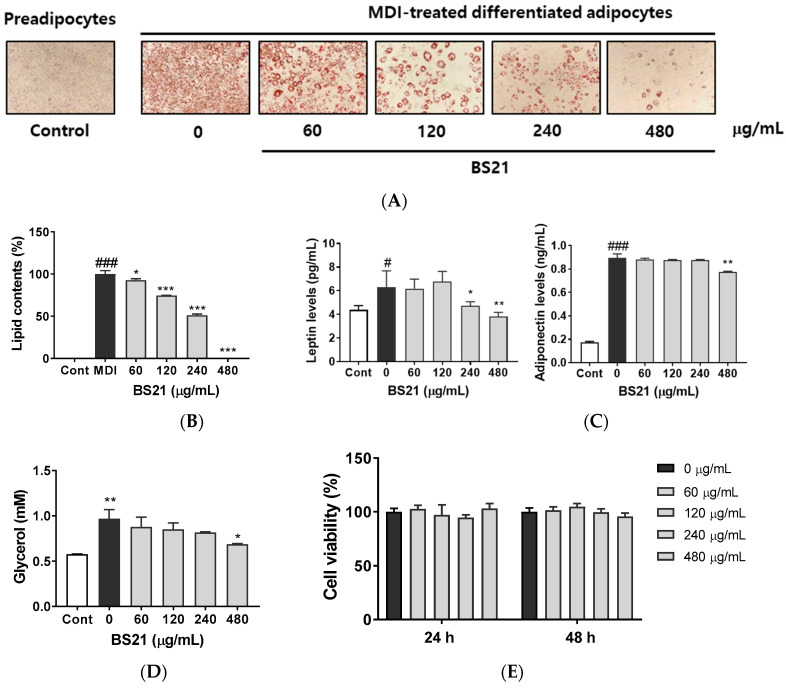
Effect of a 2:1 mixture of *Phyllostachys pubescens* and *Scutellaria baicalensis* (BS21) on adipocyte differentiation in 3T3-L1 cells. (**A**) Oil Red O staining of intracellular triglycerides in 3T3-L1 cells. 3T3-L1 cells were treated with BS21 (60, 120, 240, and 480 μg/mL) during differentiation induction. (**B**) Relative densities of lipid contents. (**C**) Leptin and adiponectin levels in cell supernatants. (**D**) Glycerol levels in differentiated cells. (**E**) Cell viability. Values are expressed as means ± SD (*n* = 3). Significant differences were observed between control (undifferentiated preadipocytes) and differentiated MDI (3-isobutyl-1-methylxanthine, dexamethasone, and insulin) cells: # *p* < 0.05, ## *p* < 0.01, and ### *p* < 0.001. Significant differences were observed between differentiated MDI cells and BS21-treated cells: * *p* < 0.05, ** *p* < 0.01, and *** *p* < 0.001.

**Figure 2 plants-09-01422-f002:**
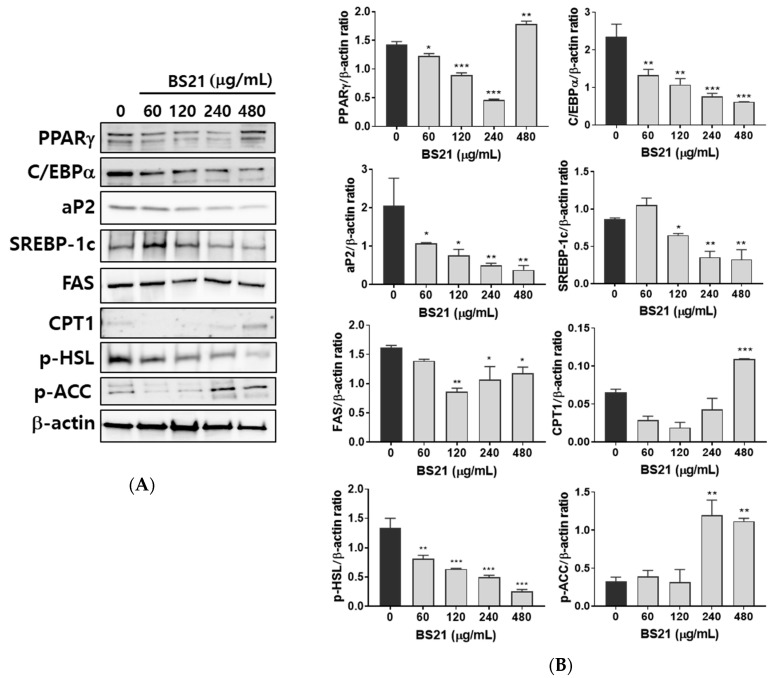
Effect of a 2:1 mixture of *Phyllostachys pubescens* and *Scutellaria baicalensis* (BS21) on lipid metabolism-related protein expression in 3T3-L1 cells. (**A**) Representative bands and (**B**) relative changes in protein expression levels. The relative expression levels of proteins were normalized against a β-actin internal control. Values are expressed as means ± SD (*n* = 3). Significant differences were observed between differentiated MDI (3-isobutyl-1-methylxanthine, dexamethasone, and insulin) cells and BS21-treated cells: * *p* < 0.05, ** *p* < 0.01, and *** *p* < 0.001.

**Figure 3 plants-09-01422-f003:**
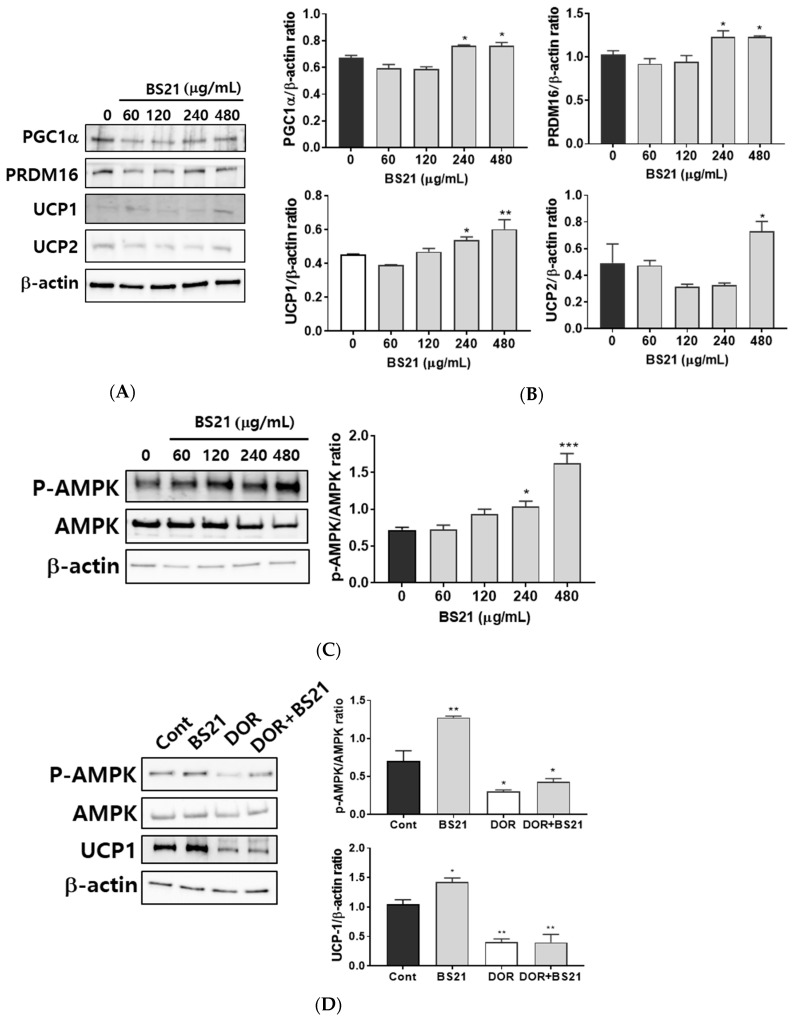
Effect of a 2:1 mixture of *Phyllostachys pubescens* and *Scutellaria baicalensis* (BS21) on brown adipocyte-specific protein expression and adenosine monophosphate-activated protein kinase (AMPK) phosphorylation in 3T3-L1 cells. (**A**) Representative bands and (**B**) relative changes in protein expression levels. The relative expression levels of proteins were normalized against a β-actin internal control. (**C**) Representative bands and relative changes in p-AMPK and AMPK protein levels are shown. Target-protein phosphorylation was normalized to the total protein-expression level. (**D**) 3T3-L1 adipocytes were pretreated with the p-AMPK antagonist dorsomorphin (5 μM) in addition to BS21 at differentiation stages. Values are expressed as mean ± SD (*n* = 3). Significant differences were observed between differentiated MDI (3-isobutyl-1-methylxanthine, dexamethasone, and insulin) cells and BS21-treated cells: * *p* < 0.05, ** *p* < 0.01, and *** *p* < 0.001.

**Figure 4 plants-09-01422-f004:**
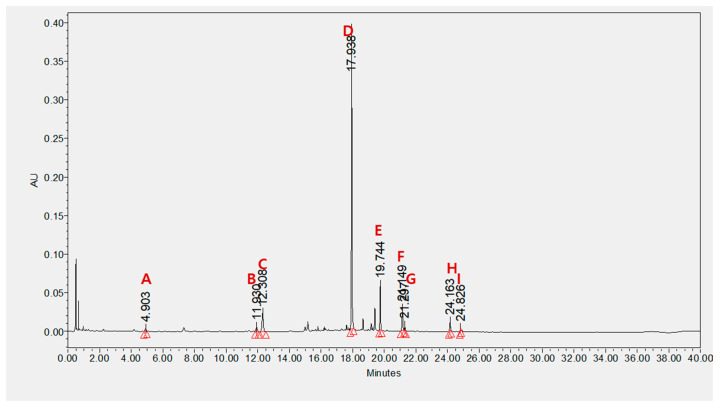
Identification and quantitative analysis of components from BS21: (**A**) cholorogenic acid, (**B**) orientin, (**C**) isoorientin, (**D**) baicalin, (**E**) wogonoside, (**F**) baicalein, (**G**) tricin, (**H**) wogonin, and (**I**) chrysin. Identification was based on comparisons of retention times and UV spectra with those of commercial standards. The components were quantified based on the peak area at 270 nm.

**Figure 5 plants-09-01422-f005:**
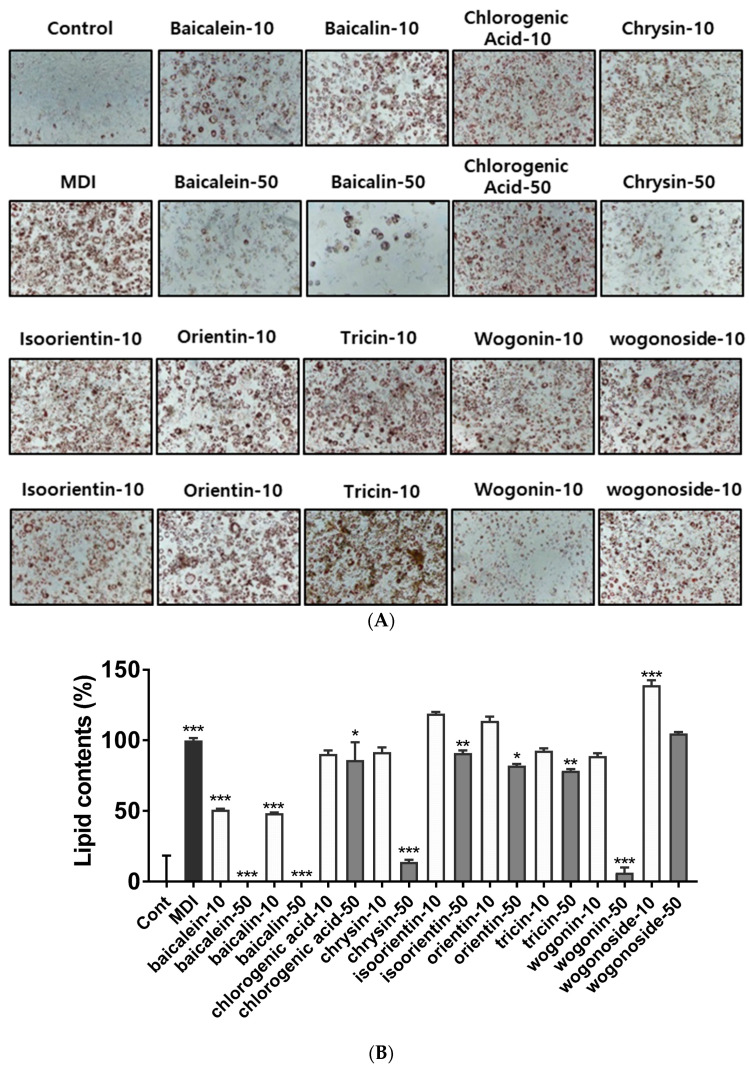
Effect of a 2:1 mixture of *Phyllostachys pubescens* and *Scutellaria baicalensis* (BS21) constituents on adipocyte differentiation in 3T3-L1 cells. (**A**) Oil Red O staining of intracellular triglycerides in 3T3-L1 cells. 3T3-L1 cells were treated with compounds (10 and 50 μg/mL) during differentiation induction. (**B**) Relative densities of lipid contents. (**C**) Leptin levels. (**D**) Cell viability. Values are expressed as means ± SD (*n* = 3). Significant differences were observed between control (undifferentiated preadipocytes) and differentiated MDI cells: # *p* < 0.05, ## *p* < 0.01, and ### *p* < 0.001. Significant differences were observed between differentiated MDI (3-isobutyl-1-methylxanthine, dexamethasone, and insulin) cells and BS21-treated cells: * *p* < 0.05, ** *p* < 0.01, and *** *p* < 0.001.

**Table 1 plants-09-01422-t001:** BS21 extract constituents.

Chlorogenic Acid	Orientin	Isoorientin	Baicalin	Wogonoside	Baicalein	Tricin	Wogonin	Chrysin
2.9 ± 0.10	2.6 ± 0.06	28.1 ± 0.35	73.2 ± 0.67	8.9 ± 0.27	4.1 ± 0.02	2.9 ± 0.50	2.2 ± 0.07	0.2 ± 0.04

Data are expressed as mean ± standard deviation (SD).

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
