# Peer review of "Herbal Combination of Phyllostachys pubescens and Scutellaria baicalensis Inhibits Adipogenesis and Promotes Browning via AMPK Activation in 3T3-L1 Adipocytes"

_plants, 2020, doi:10.3390/plants9111422_

Round 1
Reviewer 1 Report
In the present study, Sung et al. aim to evaluate the effect of BS21, a combination of of Phyllostachys pubescans and Scutellaria biocalensis extracts in on 3T3L1-adipocytes. They find BS21 affects lipid metabolism of 3T3L1-adipocytes. In particular, higher concentrations of BS21 (240 and or 480µg/ml) decreased lipid content, adiponectin and leptin levels, and as well as levels of proteins important for adipocyte maturation and function (PPARγ, CEBPα, AP2, SREBP1c) but instead increased proteins implicated in beta oxidation (CPT1) and adipocyte browning (PGC1α, PRDM16, UCP1) without affecting cell viability. Further, as increased UCP1 expression is diminished after AMPK inhibition, hey claim, that BS21 induced browning depends on AMPK phosphorylation. Next, via HPLC analysis they identified 9 active substances of BS21 which were able to inhibit 3T3L-1 differentiation as measured by lipid accumulation and leptin levels in a similar manner.
The present study aims to elucidate the mechanism for the observed anti-obesity effects of BS21 in mice published previously by the authors. While the idea of the study is reasonable, the described results rose some questions:
First of all, it is not stated how many wells (what wells) were analyzed? How many replicates/wells are indicated as one bar? How often was the study repeated? Not only in the methods section but also in every figure legend n-number should be provided.
As the authors identify AMPK activation as their central mechanism and claim that AMPK activation “switches on mitochondrial fatty acid oxidation and lipolysis “ (Discussion line 180), but do not see increased but rather decreased lipolysis after BS21 treatment (Figure 1d), they should at least discuss this discrepancy in their discussion.
Next, they measure HSL as a marker for lipolysis. Lipolysis is triggered by phosphorylated HSL (Ser660) and thus, usually, Phospho-HSL levels are considered as a measure of lipolysis.
Methodologically, the authors use UPLC analysis to identify active components of BS21. Here, it is not clear, how the substances are identified exactly? Is there retention time compared to known standards or is there a database-based identification?
To state that “BS21 dose dependently decreased….” (discussion line 157) is exaggerated. Please be more precise.
Minor comments:
The Abbreviation MDI cells should be introduced.
Reviewer 2 Report
Dear Editor,
I carefully read the article by Sung et al., which regards an overall interesting topic.
My remarks for the authors:
- Some typos need to be corrected before resubmission. Throughout the manuscript there is also a problem of verb tense concordance, that authors need to resolve before publication. In general, the paper should be revised by a native-English speaking person.
- Ref. 1 is out of context and should be replaced with a more pertinent articles (such as doi: 10.1159/000442721).
- Table 1: Authors should also specify in the caption how the parameters were expressed in the table (i.e. mean plus/minus standard deviation).
- Fig.1-3, Fig.5: All the used abbreviations should be specify in the caption of the figures, for clarity.
- Results should be better discussed in the appropriate paragraph. Furthermore, authors should comment the limitations of their study.
Round 2
Reviewer 2 Report
Dear Editor,
I carefully read the revised version of the manuscript, which is significantly improved in comparison with the previous one. I suggest the acceptation of the manuscript in the journal.
This manuscript is a resubmission of an earlier submission. The following is a list of the peer review reports and author responses from that submission.